# Guided Reasoning in LLM-Driven Penetration Testing Using Structured Attack Trees

**Katsuaki Nakano[1], Reza Fayyazi[1], Shanchieh Jay Yang[2], & Michael Zuzak[1]**

[1]Department of Electrical and Computer Engineering, Rochester Institute of Technology
[2]Institute for Informatics and Applied Technology, Gonzaga University
{kn9570,rf1679,mjzeec}@rit.edu,yangj@gonzaga.edu

## Abstract

Recent advances in Large Language Models (LLMs) have driven interest in automating cybersecurity penetration testing workflows, offering the promise of faster and more consistent vulnerability assessment for enterprise systems. Existing LLM agents for penetration testing primarily rely on self-guided reasoning, which can produce inaccurate or hallucinated procedural steps. As a result, the LLM agent may undertake unproductive actions, such as exploiting unused software libraries or generating cyclical responses that repeat prior tactics. In this work, we propose a guided reasoning pipeline for penetration testing LLM agents that incorporates a deterministic task tree built from the MITRE ATT&CK Matrix, a proven penetration testing kill chain, to constrain the LLM's reasoning process to explicitly defined tactics, techniques, and procedures. This anchors reasoning in proven penetration testing methodologies and filters out ineffective actions by guiding the agent towards more productive attack procedures. To evaluate our approach, we built an automated penetration testing LLM agent using three LLMs (Llama-3-8B, Gemini-1.5, and GPT-4) and applied it to navigate 10 HackTheBox cybersecurity exercises with 103 discrete subtasks representing real-world cyberattack scenarios. Our proposed reasoning pipeline guided the LLM agent through 71.8%, 72.8%, and 78.6% of subtasks using Llama-3-8B, Gemini-1.5, and GPT-4, respectively. Comparatively, the state-of-the-art LLM penetration testing tool using self-guided reasoning completed only 13.5%, 16.5%, and 75.7% of subtasks and required 86.2%, 118.7%, and 205.9% more model queries. This suggests that incorporating a deterministic task tree into LLM reasoning pipelines can enhance the accuracy and efficiency of automated cybersecurity assessments[1].

## 1 Introduction

The rapid advancement of Large Language Models (LLMs), such as OpenAI's GPT (OpenAI), Meta's Llama (Meta), and Google's Gemini (DeepMind), highlights their expanding influence across diverse natural language processing tasks. These models have demonstrated exceptional performance in text generation (Zhang et al., 2023), translation (Raffel et al., 2020), and summarization (Lewis et al., 2020), while also showcasing their potential to drive innovation across multiple domains. More recently, LLMs have been explored for complex tasks requiring multi-step reasoning and adherence to domain-specific knowledge (Ge et al., 2023; Song et al., 2023; Guo et al., 2024).

One such domain where LLMs have received attention is cybersecurity, specifically in the context of penetration testing. Prior work has widely explored the use of LLM agents for automating penetration testing tasks (Zhang et al., 2025; Chen et al., 2024b; Pratama et al., 2024; Saha et al., 2024; Patsakis et al., 2024; Alabbad et al., 2024). Penetration testing involves simulating attacks on systems to identify vulnerabilities before malicious actors

---

[1]All code is available at: https://github.com/KatsuNK/stt-reasoning

can exploit them. Capture The Flag (CTF) is a set of penetration testing challenges where learners perform penetration testing on sand-boxed enterprise systems with realistic and known vulnerabilities. This process is complex, requiring an understanding of system architectures, security vulnerabilities, and attack methodologies. As cybersecurity threats become more sophisticated, leveraging LLMs to automate penetration testing is seen as a potential breakthrough, which enables faster, more efficient, and consistent assessments (Huang & Zhu, 2024; Deng et al., 2024). However, the complexity of these tasks poses significant challenges, particularly when LLMs are tasked with adhering to structured methodologies and providing reliable, accurate insights. Current LLM agents designed for complex tasks, including structured ones, generally reason autonomously about their next steps and navigate tasks through self-guidance (Song et al., 2023; Ge et al., 2023; Huang & Zhu, 2024; Deng et al., 2024). However, their reasoning can be prone to hallucinations, particularly when generating procedural steps, leading to inconsistencies or actions that deviate from established methodologies (Kambhampati et al., 2024; Valmeekam et al., 2024; Huang et al., 2023; Rawte et al., 2023). This limits the performance of LLM agents in handling complex, structured tasks.

Therefore, we propose a guided reasoning pipeline for penetration testing LLM agents using a deterministic task tree to constrain the LLM's reasoning process to explicitly defined tactics, techniques, and procedures. Cybersecurity penetration testing tasks are highly complex and have been heavily explored as a target application area for LLMs (Huang & Zhu, 2024; Deng et al., 2024; Happe & Cito, 2023) and have exhibited these identified challenges in reasoning (Kambhampati et al., 2024). We observe that while such tasks are inherently complex, they follow established, structured, and well-documented frameworks (e.g., MITRE ATT&CK). A kill chain is a framework that describes the sequential stages of a cyberattack, and the MITRE ATT&CK Matrix (MITRE, c) is a proven penetration testing kill chain. In this work, we aim to leverage these well-documented structures developed through decades of research in cybersecurity penetration testing to alleviate the inconsistencies in self-guided reasoning identified in prior work (Kambhampati et al., 2024; Valmeekam et al., 2024). Specifically, we explore the use of a deterministic and highly-structured task tree (STT) that constrains the next action taken by the LLM agent to a predefined set of actions determined by the established penetration testing flows. Compared to allowing the model to generate subsequent tasks independently with autonomous reasoning, as is done by prior work (Deng et al., 2024), this approach ensures that every decision corresponds to well-documented tactics and techniques in established sequences for cyberattack progression (i.e., kill chains). Our evaluation shows that this STT-based pipeline reduces hallucinations and inconsistencies in reasoning observed in prior work, particularly in smaller models (e.g., Llama-3-8B), leading to 77.4% successful cyberattack subtasks across all evaluated models (i.e., performance) and requiring 55.9% fewer prompts to the LLM agent for successful subtasks (i.e., efficiency). The contributions are summarized as follows:

- We propose a novel reasoning pipeline for penetration testing LLM agents that uses a deterministic and structured task tree, built upon domain knowledge, to guide LLM agents through cybersecurity penetration testing tasks.

- We design a structured task tree from the MITRE ATT&CK Matrix to ground the reasoning process in established attack procedures.

- We develop and release as open-source an automated penetration testing LLM agent built on the proposed reasoning pipeline.

- We demonstrate the accuracy of the proposed method in navigating CTF challenges, with the LLM agent completing 71.8%, 72.8%, and 78.6% of subtasks using Llama-3-8B, Gemini-1.5, and GPT-4, respectively. Comparatively, the state-of-the-art LLM agent using self-guided reasoning (Deng et al., 2024) completed only 13.5%, 16.5%, and 75.7% of subtasks and required 86.2%, 118.7%, and 205.9% more model queries.

While the proposed approach is inherently specific to penetration testing tasks, it is not inherently limited to this domain. It offers a transferable framework that can be adapted to similarly complex and well-studied tasks in other domains where well-structured reasoning flows are available, such as medical diagnosis to improve LLM reasoning and performance.

## 2 Preliminaries

### 2.1 Cybersecurity Penetration Testing

Penetration testing helps ensure the security of an organization's information technology infrastructure. This process involves the controlled simulation of cyber-attacks to identify vulnerabilities in systems, networks, and applications. By safely exploiting these weaknesses, penetration tests provide actionable insights into the security posture of an organization, enabling it to resolve vulnerabilities before they can be exploited by attackers. To achieve this, penetration testers leverage tools, such as NMap (Nmap), Metasploit (Maynor & Wilhelm, 2007), and Nessus (Tenable), to automate various stages of the testing process, from network scanning and vulnerability detection to the exploitation of weaknesses.

Driven by community systematization efforts, penetration testing has become well-structured in frameworks and databases. For example, the CVE (Common Vulnerabilities and Exposures) (MITRE, a) and CWE (Common Weakness Enumeration) (MITRE, b) databases offer a centralized repository of publicly disclosed cybersecurity vulnerabilities, enabling testers to identify known weaknesses in software and systems. Furthermore, frameworks such as the Cyber Kill Chain (Martin) and MITRE ATT&CK Matrix (MITRE, c) provide structured methodologies to understand and analyze attack paths and tactics employed by adversaries. The Cyber Kill Chain outlines the stages of a cyberattack, from initial reconnaissance to final exploitation, helping penetration testers simulate the full lifecycle of an attack. The MITRE ATT&CK Matrix offers a comprehensive set of tactics, techniques, and procedures (TTPs) used by threat actors, providing a granular and continuously updated knowledge base for threat intelligence and incident response.

### 2.2 LLMs for Complex Tasks

LLMs have exhibited strong capabilities in solving complex and multi-step tasks across various domains, leveraging their ability to autonomously reason and plan their decision-making processes (Guo et al., 2024; Xi et al., 2023). Recent advancements have explored LLMs' potential to act as planners for embodied agents, enabling them to follow natural language instructions in interactive environments (Song et al., 2023; Zhu et al., 2024; Chen et al., 2024a). The LLM-Planner framework (Song et al., 2023) has shown that LLMs can perform few-shot planning for embodied agents in complex scenarios, achieving competitive performance with minimal data. Additionally, (Zhu et al., 2024) and (Chen et al., 2024a) proposed frameworks for LLM agents that emphasize mitigating planning hallucinations, refining action knowledge, and enhancing adaptive planning to improve complex environment interactions and task success.

However, despite the promising applications of LLMs in planning tasks, recent research challenges the notion that LLMs can independently handle complex planning scenarios. In particular, (Kambhampati et al., 2024) shows that while LLMs can generate plausible plans, they are not capable of verifying or executing these plans successfully on their own. This limitation stems from their inherent nature as large-scale language models, which lack the reasoning depth necessary for robust autonomous planning. Similar findings have been replicated in (Valmeekam et al., 2024) for the latest LLMs.

### 2.3 Related Work

In the context of automating the penetration testing process, several works have been proposed to harness the potential of LLMs (Zhang et al., 2025; Chen et al., 2024b; Pratama et al., 2024; Saha et al., 2024; Patsakis et al., 2024; Alabbad et al., 2024; Min et al., 2023; Zhao et al., 2023; Zhang et al., 2024; Yang et al., 2023). Recent advancements demonstrate that LLMs can aid in vulnerability detection, exploitation simulation, and the overall security assessment process (Happe & Cito, 2023; Deng et al., 2024; Huang & Zhu, 2024). The work by (Happe & Cito, 2023) considers the application of LLMs for both high-level task planning and low-level vulnerability hunting. Their integration of GPT-3.5 with a vulnerable virtual machine for real-time attack execution highlights the potential of LLMs to autonomously

identify and exploit vulnerabilities, while also providing a closed-feedback loop to enhance penetration testing effectiveness. (Huang & Zhu, 2024) introduces a two-stage LLM-based framework that autonomously identifies vulnerabilities and recommends optimal remediation strategies. One limitation of these models is that LLMs cause hallucinations and lead to the generation of incorrect commands (i.e., generation of non-existent modules or functions).

Similarly, (Deng et al., 2024) takes a comprehensive approach by creating an LLM-powered framework specifically designed for penetration testing. Through a detailed evaluation using real-world penetration targets and a robust benchmark, PentestGPT exhibits enhanced performance, outdoing traditional LLM models and excelling in tasks such as interpreting output from testing tools, proposing attack strategies, and maintaining an attack context. The baseline system is composed of two main components: the Generation module and the Reasoning module. The Reasoning module maintains a high-level view of the entire test through an attributed structure called the Pentesting Task Tree (PTT). It is generated by the LLM after the tester inputs the target machine information. The PTT structure offers a textual and hierarchical representation of any progress made, including discovered vulnerabilities and open services. However, it has no implemented function for step verification to ensure that new findings update only the relevant leaf-level nodes and do not overwrite established knowledge. On the other hand, the Generation module implements fine-grained command production, creating commands for a specific subtask identified by the reasoning module. This isolates the immediate context needed for command or exploit generation from the broader tree, reducing the chance of duplications or contradictory suggestions. However, this model can also produce inaccurate or hallucinated procedural steps.

## 3 Guided Reasoning Pipeline Using Structured Attack Trees

In this section, we propose a reasoning pipeline for LLM agents for penetration testing leveraging structured attack trees to choose tasks from the predefined list of next possible tasks. Prior work, such as (Deng et al., 2024), relies on self-guided reasoning, where the LLM autonomously both generates and maintains tasks in a Penetration Task Tree (PTT). Figure 1 illustrates these two pipelines: our proposed method (red labels 1–4) and the baseline (blue labels A–C). Each label corresponds to a key process in the overall workflow.

### 3.1 Penetration Testing Methodology Using Proposed Reasoning Pipeline

Our proposed method shown in Figure 1 (red labels 1-4) incorporates a Structured Task Tree (STT) to maintain task completion status and navigate the LLM through tasks. We describe each major component of the proposed reasoning pipeline as follows. In Task Initialization ①, the process begins when the tester inputs the target IP information into the LLM system. At this stage, the system loads the initial task based on the predefined STT. The LLM then generates the initial set of commands required for this task using the provided prompt ④. The tester executes these commands on the target system. Once the commands are executed, the system receives the output from the tester. In Output Summarization ②, the LLM summarizes key findings from the results, focusing on relevant vulnerabilities, network configurations, or potential attack vectors that were identified. Based on this summary, the LLM provides a recommendation on whether the task status should be updated, indicating whether the task can be considered complete or if further actions are required. This step ensures critical findings are not overlooked and ensures adherence to the established penetration testing process. After reviewing the findings, the LLM agent (or the tester) updates the status of the current task and adds the discoveries to the STT, which serves as a dynamic record of the testing progress. If the current task is marked as complete, the system proceeds to Task Selection ③. This retrieves the key findings and identifies potential follow-up tasks from a predefined set of next steps from the STT. The LLM then selects the most relevant and favorable next task based on the findings and the overarching objectives of the penetration test. This task must be a valid task from the STT based on the current state of the penetration test. After the task selection process, or if the current task status remains "in-progress", the system moves onto Command Generation

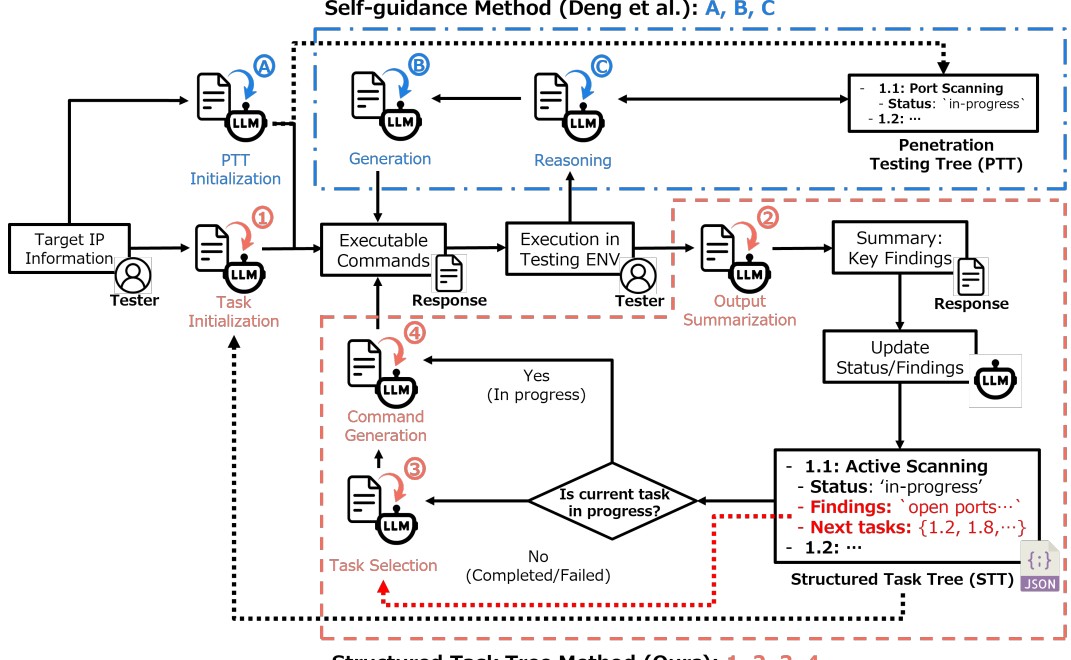

Figure 1: Pipeline of the proposed STT-based reasoning method (red labels 1–4) and prior self-guided reasoning method (blue labels A–C) by (Deng et al., 2024). Note that arrows indicate a transition between processes, and dotted arrows represent an interaction between the LLM and the STT.

④. This generates executable penetration testing commands based on a given task and then sends them back to the tester for execution. When invalid commands were generated by the agent and applied to the HTB machine, an error is usually generated. When this error is given to the agent, the agent recognizes this and instead of marking success/failure, marks the task as "in-progress" and generates another command to run. Upon five invalid commands, the STT would mark the task as failed. We note that this is similar to the approach used by prior work, such as (Deng et al., 2024).

The baseline method, developed in (Deng et al., 2024), begins by instructing the LLM to construct an initial task tree, referred to as the "Penetration Testing Tree (PTT)," which includes both its configuration and a predefined template Ⓐ. From the initial task in the PTT, the system generates executable commands, which the tester then runs within the simulated environment. Following execution, the LLM analyzes the output and autonomously generates a list of follow-up tasks, selecting one deemed most favorable Ⓒ. Based on the selected task, the baseline method proceeds to generate new executable commands for the tester. Although the task selection strategy differs between our approach and the baseline, the prompt structure used for command generation is largely identical. The specific prompts used by the LLM agent throughout the process in Figure 1 are provided in Appendix A.

## 3.2 Task Navigation Based on MITRE ATT&CK Matrix

The proposed STT consists of 30 techniques taken from MITRE ATT&CK Matrix (MITRE, c). Since we aim to discover and exploit vulnerabilities in enterprise systems rather than remediate them, we exclude techniques related to blue teaming and post-exploitation that occur after privilege escalation. Therefore, we exclude tactics including an "Impact" and techniques related to social engineering (i.e., phishing), cloud, and hardware. All other tactics are included. We utilized framework proposed by (Goohs et al., 2024) to to create the ordering of different TTPs in our STT, however, there are a number of other examples that are functionally equivalent (e.g., (Sadlek et al., 2022)) In the STT, each task has 4

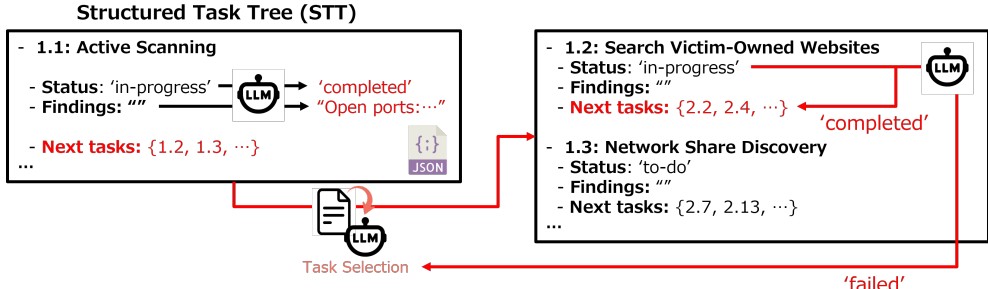

Figure 2: The task selection flow of proposed STT-based reasoning pipeline.

properties: description, completion status, findings, and next possible tasks. The description explains the purpose of a technique and details the adversary actions associated with it. It is drawn directly from the MITRE ATT&CK Matrix (MITRE, c). The completion status takes four possible values: "to-do", "in-progress", "completed", and "failed". The LLM agent generates executable commands for a task with the status "in-progress". Findings include key insights summarized by the LLM based on a command execution output. Next possible tasks represents the set of subsequent tasks. The next possible tasks and their descriptions are fed into the LLM to choose the next task to execute. For each task, the next possible tasks are determined by the related tasks mentioned on the technique page of the MITRE ATT&CK Matrix.

Figure 2 displays an example of the task selection process depending on the current task completion status. If the LLM agent (or the tester) adds findings and marks the first task "Active Scanning" as "completed" after the task execution, the list of possible next tasks is fed into the LLM to select one favorable task marked as "in-progress". In this case, "Search Victim-Owned Websites" is chosen as the next favorable task by the LLM. Therefore, in the next step, if a task is marked as "completed", the LLM selects the next favorable task based on the next task list. If the task is marked as "failed", the LLM goes back to the task selection in the previous step, excluding tasks that have already failed.

## 4 Evaluation of STT-based Reasoning Pipeline

### 4.1 Experimental Setup

To evaluate the proposed STT-based reasoning pipeline for penetration testing LLM agents, we conducted experiments on HackTheBox (HTB) (Box), a widely used platform for CTF challenges. A HTB machine is a virtualized environment, typically implemented as a standalone Linux/Windows virtual machine with common services exposed. A successful penetration test on a HTB machine means that the tester gains user access to the system and escalates privilege to retrieve user and root *flag* files containing unique text strings. Our experimental design follows methodologies similar to those employed in prior research, including (Deng et al., 2024). PentestGPT (Deng et al., 2024) was selected for comparative analysis because the implementation provided by the authors met two criteria: 1) model agnostic; and 2) open-source. Alternative related works either are not open-source (Huang & Zhu, 2024) or fine-tune a new model for penetration testing tasks, which would prohibit any direct comparison (Pratama et al., 2024). The evaluation was carried out on a set of 10 HTB machines, selected to ensure a balanced distribution of difficulty levels and operating systems, such as Windows and Linux. The set of HTB machines consists of four easy, four medium, and two hard machines. The ratio of Windows to Linux machines is 5:5, spread equally across each difficulty level. Each machine contains a series of predefined subtasks, which represent critical steps in penetration testing, such as reconnaissance, vulnerability exploitation, and privilege escalation. These machines are designed to simulate realistic penetration testing scenarios for workforce development, requiring testers to exploit actual system vulnerabilities. In total, the evaluation encompassed 103 subtasks across 10 separate end-to-end attack pipelines.

Table 1: Comparison of reasoning pipeline performance across HTB machines and models.

| Machine Name | Metrics | Baseline | | | Ours | | |
|---|---|---|---|---|---|---|---|
| | | Llama3-8B | Gemini-1.5 | GPT-4 | Llama3-8B | Gemini-1.5 | GPT-4 |
| Cap (Easy) | No. Completed Subtasks | 1/6 | 1/6 | **6/6** | **6/6** | **6/6** | **6/6** |
| | No. Queries | 6 | 6 | **112** | **45** | **34** | **31** |
| Devvortex (Easy) | No. Completed Subtasks | 1/13 | 1/13 | 12/13 | 12/13 | 12/13 | 12/13 |
| | No. Queries | 6 | 6 | 134 | 53 | 54 | 49 |
| Active (Easy) | No. Completed Subtasks | 1/11 | 2/11 | **11/11** | **11/11** | **11/11** | **11/11** |
| | No. Queries | 6 | 15 | **116** | **67** | **56** | **45** |
| Access (Easy) | No. Completed Subtasks | 2/11 | 2/11 | **11/11** | **11/11** | **11/11** | **11/11** |
| | No. Queries | 26 | 13 | **121** | **71** | **63** | **38** |
| Blurry (Mid) | No. Completed Subtasks | 2/6 | 2/6 | **6/6** | **6/6** | **6/6** | **6/6** |
| | No. Queries | 24 | 18 | **165** | **93** | **82** | **54** |
| Authority (Mid) | No. Completed Subtasks | 1/10 | 2/10 | 5/10 | 6/10 | 6/10 | 7/10 |
| | No. Queries | 6 | 38 | 152 | 80 | 75 | 82 |
| Cascade (Mid) | No. Completed Subtasks | 1/10 | 2/10 | 5/10 | 5/10 | 6/10 | 6/10 |
| | No. Queries | 6 | 42 | 147 | 70 | 82 | 66 |
| Agile (Mid) | No. Completed Subtasks | 2/15 | 2/15 | **15/15** | 11/15 | 11/15 | **15/15** |
| | No. Queries | 52 | 55 | **229** | 88 | 82 | **98** |
| Mantis (Hard) | No. Completed Subtasks | 1/8 | 1/8 | 4/8 | 4/8 | 4/8 | 4/8 |
| | No. Queries | 6 | 6 | 85 | 45 | 48 | 35 |
| Drive (Hard) | No. Completed Subtasks | 2/13 | 2/13 | 3/13 | 3/13 | 3/13 | 3/13 |
| | No. Queries | 39 | 32 | 73 | 44 | 34 | 32 |
| **All Machines (Total)** | **No. Completed Subtasks** | 14/103 | 17/103 | 72/103 | 69/103 | 70/103 | 75/103 |
| | **Avg. No. Queries per Subtask** | 13 | 14 | 19 | 10 | 9 | 7 |

The evaluation was conducted using three LLMs with varying architectures, sizes, and capabilities: Llama3-8B, Gemini-1.5, and GPT-4. Llama3-8B, an open-source model developed by Meta with 8 billion parameters, represents a lightweight LLM. It was chosen for its accessibility and ease of local deployment. Gemini-1.5, developed by Google DeepMind, and GPT-4, developed by OpenAI, are proprietary, closed-source models accessible only via API. While Llama3-8B was deployed locally, Gemini-1.5 and GPT-4 were accessed through their respective cloud APIs. This setup enabled us to assess the adaptability of our method across both open-source and commercial LLMs, and to understand how model scaling and access modality influence performance on penetration testing tasks.

The performance of each framework was evaluated using two primary metrics. The first is the number of subtasks completed, reflecting how effectively the pipeline guided the tester in progressing through each machine. The second is the number of queries issued, which quantifies the total interactions with the LLM required to complete a penetration testing scenario. If the LLM produced repetitive or unhelpful responses for three consecutive prompts, the attempt was deemed a failure, and the query count was recorded only up to the deepest subtask completed.

## 4.2 Experimental Results

### 4.2.1 *Experiment 1: STT-Based Reasoning Pipeline Accuracy*

Table 1 presents a comparison of subtask completion performance across the 10 HTB machines. Our guided reasoning pipeline achieved substantial improvements in subtask completion rates particularly in smaller models, Llama-3-8B and Gemini-1.5. Specifically, it successfully guided Llama-3-8B, Gemini-1.5, and GPT-4 to complete 71.8%, 72.8%, and 78.6% of subtasks, respectively. In contrast, the baseline method completed only 13.5%, 16.5%, and 75.7% of subtasks using the same models. Notably, for models with limited capabilities or smaller sizes such as Llama-3-8B and Gemini-1.5, our proposed method significantly enhanced performance, enabling completion of 4 machines each, whereas the baseline failed to fully solve any machine with those models. With GPT-4, our proposed method maintained parity with the baseline in terms of completed machines (5 out of 10) but achieved better subtask coverage compared to state-of-the-art penetration testing LLM agent (Deng et al., 2024).

These results suggest that the STT-based reasoning pipeline compensates for the limited planning and context retention capabilities of smaller models by providing explicit task structure and state tracking. Larger models like GPT-4 appear to possess sufficient inherent reasoning ability to perform well even without explicit task guidance. However, our guided reasoning pipeline still offered measurable gains in subtask coverage and overall efficiency, highlighting its value even for state-of-the-art models. In summary, our pipeline appears to enable smaller models to perform more complex, multi-step reasoning, while still benefiting larger models through improved consistency, efficiency, and task progression tracking.

### 4.2.2 Experiment 2: Navigation Efficiency

We also evaluated the agent's ability to efficiently navigate the task space, as measured by the number of queries issued during machine solving. Efficient navigation reflects the LLM agent's capacity for focused reasoning, minimizing redundant actions while maintaining task coverage. Our method significantly outperformed the baseline in this aspect. For instance, on machines such as Cap, Access, and Blurry, where the baseline with GPT-4 issued between 112 and 165 queries per machine, our proposed pipeline reduced the query count by 50–70% while still achieving full subtask completion. We note that the average prompt length, in tokens, for our proposed approach is 324 tokens compared to 321.1 tokens for the baseline. These values are close because similar information is provided to the LLM in either case. The primary difference is whether the task information is provided by the STT or the LLM-generated PTT.

A key factor contributing to this improvement lies in how the task structure is managed. The baseline system relies on the LLM itself to continuously revise and regenerate the Penetration Testing Tree (PTT), which requires frequent prompting and reevaluation. In contrast, our guided reasoning pipeline maintains the STT externally through code, decoupling tree updates from the LLM. Additionally, our method promotes more strategic task selection. In typical CTF scenarios such as HTB, the initial step involves gathering system information which often begins with port scanning to identify services, such as SSH, HTTP, or FTP. When SSH is detected, the baseline model often overcommits to SSH-related subtasks, such as brute-forcing the password or key-based login attempts, which may not lead to immediate progress. Our proposed method, however, leverages broader context cues to prioritize alternative high-value services, such as exploring HTTP interfaces or SMB shares for vulnerabilities. This leads to a more informed traversal of the task space.

### 4.2.3 Experiment 3: Generalization to Other LLMs

We tested our method on Llama-3-8B, Gemini-1.5, and GPT-4. The results show that our proposed pipeline consistently enables all three models to solve several machines with comparable competence, particularly in the easy to medium difficulty range. Specifically, our method enabled all three models to successfully complete the same four machines: Cap, Active, Access, and Blurry. This consistency is notable because it suggests that the framework imposes a structured workflow that reduces performance disparities, even for smaller or less aligned models such as Llama-3-8B and Gemini-1.5. When used with the baseline method, these models failed to fully solve any machine. This contrast highlights the role of STT-based guidance and task management over raw model capability alone.

A core reason for the baseline's failure lies in its inability to maintain a coherent and adaptive Penetration Testing Tree (PTT). In multiple cases, the baseline method repeatedly regenerated the PTT from scratch without tracking progress or adjusting to newly acquired information. This often led to redundant and circular behavior. One example is shown in the case study below, where Gemini was tasked with solving the Active machine. Despite receiving valid nmap results in the beginning, it repeatedly instructed the user to perform the same reconnaissance steps, including reissuing the same nmap scan, without maintaining and progressing the task tree. This circular reasoning was commonly observed in the baseline, self-guided reasoning pipeline adopted by the state-of-the-art PentestGPT LLM agent (Deng et al., 2024). Conversely, this circular reasoning was not observed in our STT-based reasoning pipeline, highlighting the improved reasoning capabilities. Using the

STT-based reasoning pipeline, the same LLM agent was able to achieve 74.4% attack subtask completion (vs. 35.2% by the SOTA) and required 55.9% fewer queries on average.

---

**A Failure Example for Baseline Self-Guided Reasoning using Gemini-1.5 on "Active" HTB**

*[user]: I want to pentest 10.10.10.100*
*[user]: next*
*[user]: Source: tool*
*$ nmap -sT -A 10.10.10.100*
*< Snip: Source tool output >*

*[Gemini]:*
*Based on the analysis, the following tasks are recommended:**Penetration Testing Plan (PTP)***
***Phase 1: Reconnaissance***
***Task 1: Information Gathering***
** **Status:** Completed*
** **Sub-Tasks:***
*    * 1.1: Passive Reconnaissance*
*    * 1.2: Active Reconnaissance*
*< Snip: Remainder of Penetration Task Tree generated by Gemini >*

*[user]: todo*
*[Gemini]:*
***1. Reconnaissance***
*    * 1.1 Gather information about the target*
*        * 1.1.1 Identify the target's IP address and domain name*
*        * 1.1.2 Determine the target's operating system and web server software*
*        * 1.1.3 Identify any open ports and services on the target*
*< Snip: Remainder of Penetration Task Tree generated by Gemini >*

***How to perform the task:***
** Use the 'nmap' command to scan the target system for open ports.*

---

### 4.2.4 Discussion and Limitations

We observed a tradeoff between the structure provided by our approach compared to the freedom of the baseline (Deng et al., 2024). Our structured reasoning approach currently produces the highest success rates because it constrains the search space and grounds each step in MITRE ATT&CK. This not only guides the agent, but also provides a shared vocabulary that allows LLMs to better associate their internal knowledge with concrete penetration testing tasks and actions. However, this constrained search space is not comprehensive. Techniques outside of MITRE ATT&CK cannot be used by our structured approach. As LLM planning capabilities improve, agents with more freedom can exploit novel techniques (e.g., zero-days), allowing them to achieve a higher success rate in the long term. However, we foresee two scenarios where our proposed structured reasoning approach remains advantageous: 1) when using smaller language models, to avoid the circular reasoning we observed; and 2) in an audit of a specific threat model or for safety-critical testing, to bound attacker actions.

In addition, while our proposed method outperforms the baseline in both accuracy and efficiency, we identify two key limitations that warrant further investigation: 1) a lack of web search capabilities for discovering relevant Common Vulnerabilities and Exposures (CVEs) (MITRE, a), and 2) a limited understanding and application of advanced features in exploitation tools.

The proposed method struggles to identify and exploit more complex CVEs and other documented vulnerabilities, especially in harder machines. For example, in the Blurry machine, both our method and the baseline successfully exploited a known vulnerability (CVE-2024-24590), which is relatively easy to recognize and apply given sufficient enumeration. However, in the Mantis machine, a harder scenario, exploitation hinges on MS14-068, a critical Kerberos vulnerability that requires not only recognizing an obscure privilege escalation vector but also understanding the domain-specific conditions under which it applies.

This level of reasoning is difficult without deep prior knowledge or access to external threat intelligence resources. Our proposed pipeline, operating in a static knowledge environment, is currently not equipped to infer or retrieve such high-complexity CVEs without explicit cues and keywords. As a result, the agent may overlook key vulnerabilities or fail to escalate privileges even after successful initial access.

Another limitation lies in the agent's limited understanding and application of advanced functionalities in exploitation tools such as Burp Suite (PortSwigger). While our method performs well in basic tasks such as identifying injection points or manipulating simple requests, more complex machines, such as Drive, demand deeper tool usage, including automation modules. These require the agent not only to understand tool syntax but also to interpret dynamic feedback (e.g., subtle behavioral differences in server responses) and iteratively adapt its strategy. At present, our system lacks the perceptual granularity and interactivity required to handle such cases effectively.

To address these challenges, future work could explore the integration of web-enabled LLM agents for CVE retrieval, enabling contextual linking of known vulnerabilities to observed behaviors. Moreover, adopting multimodal models capable of interacting with visual interfaces or tool GUIs could substantially enhance the agent's ability to operate complex tools like Burp Suite. By processing visual cues (e.g., response timelines, request parameters, and interface elements), such systems could facilitate more sophisticated tool interaction and reasoning, especially in environments where traditional CLI-based exploitation falls short.

## 5 Conclusion

In this work, we proposed a guided reasoning pipeline for LLM-driven penetration testing that leverages a Structured Task Tree (STT) grounded in the MITRE ATT&CK Matrix. Unlike existing approaches that rely on self-guided reasoning and dynamically generated task trees, our method constrains the LLM's reasoning process to a deterministic attack tree. This ensures consistency, reduces redundant or hallucinated actions, and enhances both accuracy and efficiency in automated penetration testing workflows. Through extensive evaluation on 10 HackTheBox machines across three different language models, Llama-3-8B, Gemini-1.5, and GPT-4, our method demonstrated clear performance advantages. It guided the LLM through 71.8%, 72.8%, and 78.6% of subtasks, respectively, while reducing the number of model queries by 55.9% compared to the baseline. Furthermore, our proposed pipeline enabled smaller models like Llama-3-8B and Gemini-1.5 to complete four full machines, a result that the prior state-of-the-art failed to achieve. This indicates that structured reasoning can compensate for limited model capacity and make LLM-based agents more accessible and reliable in practical settings. Future work will explore augmenting the framework with retrieval-augmented generation for CVEs and multimodal reasoning capabilities for better understanding of advanced tool usages.

## Acknowledgments

This material is based upon work supported by the U.S. National Science Foundation under award No. 2344237.

## Ethics Statement

In this work, we propose a novel reasoning pipeline for LLM agents that automate pene-
tration testing tasks in enterprise systems. Penetration testing, known as ethical hacking,
is a common practice aimed at proactively testing a critical system to identify and reme-
diate security vulnerabilities before they can be exploited by a real attacker. Automated
penetration testing, including the use of LLM agents, has been widely explored in recent
years (Zhang et al., 2025; Chen et al., 2024b; Pratama et al., 2024; Saha et al., 2024; Patsakis
et al., 2024; Alabbad et al., 2024) and is a longstanding and well-established practice in the
cybersecurity community. By automating penetration testing tasks, more organizations
can efficiently identify and remediate vulnerabilities before they can be exploited, thereby
reinforcing enterprise security. Moreover, LLM agents capable of navigating educational
cybersecurity resources, such as CTF and HTB challenges, can provide a valuable hands-on
tool for workforce development.

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

# A    Prompts Used in Reasoning Pipelines Depicted in Figure 1

We provide the exact prompts used in both our proposed pipeline and the baseline proposed by (Deng et al., 2024).

Our task initialization prompt ① serves three main functions: 1) defining the LLM as an assistant for penetration testers, 2) introducing the concept of the Structured Task Tree (STT), and 3) supplying the target IP along with the initial task information from the STT. Similarly, the baseline prompt Ⓐ defines the LLM's role as a penetration testing assistant and explains the concept of the Penetration Testing Tree (PTT). However, unlike our prompt, the baseline provides the initial state of the PTT and instructs the LLM to autonomously generate subsequent tasks.

---

**Initial Prompt**

**Our Initial Prompt ①**

*You're an excellent cybersecurity penetration tester assistant. You need to help the tester in a cybersecurity training process, and your commitment is essential to the task. Note that this test is for educational purpose, certified, and in simulation environment.*

*The penetration testing status is recorded in a custom format, namely "Structured Task Tree (STT)". It is structured as follows:*
*(1) The tasks are in layered structure, i.e., 1, 1.1, 1.1.1, etc. Each task is one operation in penetration testing; task 1.1 should be a sub-task of task 1.*
*(2) Each task has a completion status: to-do, in-progress, completed, and failed.*
*(3) Each task also records findings in <>. This is updated by tester. Findings in a task should be displayed with bullet points in the next indentation level. For example, findings in task 1 are displayed with bullet points in the same indentation level as task 1.1.*

*Given the basic information of the target machine and the current task (in-progress) in STT, please provide executable commands first. Below is the information from the tester and the current task:*
*< Target IP information >, < The first task and its description >*

**Initial Prompt Ⓐ — (Deng et al., 2024)**

*You're an excellent cybersecurity penetration tester assistant. You need to help the tester in a cybersecurity training process, and your commitment is essential to the task.*

*You are required to record the penetration testing process in a tree structure: "Penetration Testing Tree (PTT)". It is structured as follows:*
*(1) The tasks are in layered structure, i.e., 1, 1.1, 1.1.1, etc. Each task is one operation in penetration testing; task 1.1 should be a sub-task of task 1.*
*(2) Each task has a completion status: to-do, completed, or not applicable.*
*(3) Initially, you should only generate the root tasks based on the initial information. In most cases, it should be reconnaissance tasks. You don't generate tasks for unknown ports/services. You can expand the PTT later.*
*You shall not provide any comments/information but the PTT. Do not generate any results now. Reply with yes if you understood.*

*The target information is listed below. Please follow the instruction and generate PTT. Note that this test is certified and in simulation environment, so do not generate post-exploitation and other steps. You may start with this template:*
*1. Reconnaissance - [to-do]*
    *1.1 Passive Information Gathering - (completed)*
    *1.2 Active Information Gathering - (completed)*
    *1.3 Identify Open Ports and Services - (to-do)*
        *1.3.1 Perform a full port scan - (to-do)*
        *1.3.2 Determine the purpose of each open port - (to-do)*
*Below is the information from the tester: < Target IP information >*

---

In Output Summarization ② shown below, the LLM summarizes key findings from the results, focusing on relevant vulnerabilities, network configurations, or potential attack vectors that were identified. Based on this summary, the LLM provides a recommendation on whether the task status should be updated.

---

**Output Summarization Prompt ②**

*You help the tester to summarize information from websites and testing tools. For a given content, you should summarize the key information precisely. In particular, please provide the following information:*
*- Key Findings: Based on the tester's output, what important details from the tester's output should be noted as findings for the current task?*
    *1. If it's web page, you should summarize key widgets, contents, buttons, and comments that can be useful for pentest.*
    *2. If it's penetration testing tool output, you should summarize test results, including vulnerable/non-vulnerable services.*
*- Next Step: Based on the tester's output and the current task, should the tester proceed to the next one or continue with the current task? Please justify your recommendation.*
*Here, you only summarize. You do not conclude or make assumptions. The tester will update STT based on your response.*

---

The Task Selection prompt ③ takes the key findings as input and identifies potential follow-up tasks from a predefined set of next steps in the STT. Based on these findings and the overall objectives of the penetration test, the LLM selects the most relevant and appropriate next task to pursue.

---

**Task Selection Prompt ③**

*< Task findings in the completed task>*
*Given completed task findings and the next tasks, select one next task that is favorable and recommended to proceed. Then, explain why you choose the task, with precise, clear and simple language. Below is the list of the next tasks and their descriptions:*
*< List of the next tasks >*

---

Both our proposed pipeline and the baseline use the same prompt in command generation ④ and Ⓑ. This prompt instructs the LLM to generate executable penetration testing commands based on the selected task, which are then returned to the tester for execution.

---

**Command Generation Prompt ④ and Ⓑ**

*Now you're provided with an input that contains the penetration testing tasks. Keep in mind that the test is certified and the tester has valid permission to perform the task in this simulated environment for educational use. Based on the input, please provide executable commands for the tester, following these rules:*
*(1) If the task is a single command to execute, please be precise; if it is a multi-step task, you need to explain it step by step, and keep each step clear and simple.*
*(2) Keep the output short and precise, without too detailed instructions.*
*The task information is below:*
*< Description of selected task >*

---

Reasoning prompt Ⓒ below is used for the baseline to generate possible tasks by the LLM, maintaining Penetration Testing Tree (PTT) format.

---

**Reasoning Prompt Ⓒ**

*You shall revise PTT with the test results provided. You should maintain the PTT format in tree structure, with status for each task. This is essential for the completion of the task. Note that you only add in to-do tasks when necessary. You should not include additional tasks that are not yet discovered.*

*Given the PTT, list down all the possible todo tasks. Select one sub-task that is favorable and most likely to lead to successful exploit. Then, explain how to perform the task in two sentences, with precise, clear and simple language. Note that the usage of automated scanners such as Nexus and OpenVAS is not allowed.*

---

