# OpenReview forum: "Guided Reasoning in LLM-Driven Penetration Testing Using Structured Attack Trees"
_colmweb.org/COLM/2025/Conference — COLM 2025_

### Official Review · Reviewer_j49b · 2025-05-09

**Rating:** 7
**Confidence:** 3
**Ethics Flag:** 1

**Summary:**

This paper introduces a new method for using LMs to perform penetration testing.
Motivated by LMs’ relative subpar planning capabilities, the authors propose designing a pipeline to guide an LM’s reasoning and actions when carrying out cybersecurity testing tasks.
Specifically, instead of putting the onus of planning on the LM, the burden is partially lifted by providing LMs a framework for selecting the appropriate tasks, rather than having to come up with the tasks themselves.
Using existing, established frameworks (e.g. MITRE ATT&CK), the authors devise a new framework that (1) adds more reflection about the findings for each task, and (2) prompts an LM to ask a discriminator and select a task rather than generate/reason about what to do next.
The authors demonstrate improved performance on 10 HackTheBox tasks/machines, evaluating three LMs.
In addition to a higher success rate, the authors also show how their framework enables more efficient task completion as well.

**Questions To Authors:**

- Minor nit: Perhaps you want to be using the “citep” instead of “cite” command for your citations, as that will add parentheses around the citations.
- Figure 1: What are the dotted arrows suggesting? By the way, nice job, I liked this figure.
- Figure 1: Curious, in the structure you’ve provided, is it possible to “rollback” a task? My interpretation is “no” - if that’s correct, would there be a good reason to incorporate “undoing”?
- Section 3.1: It sounds like the pool of tasks that are available to the LM to select is a finite set. If that is correct, how comprehensive are these tasks in terms of its coverage of penetration testing settings in general? Is there potentially a pentest scenario where new tasks must be added for successful completion?
- Lines 202-212: This paragraph I found interesting in terms of contrasting it with prior work. Is there a tradeoff between the structure provided by your approach vs. the freedom of Deng et al. (2024)? In the long run, I’m curious if you think an LM’s planning capabilities will be good enough that at some point, it is preferable to give the LM more freedom, rather than having it subscribe to a particular task scaffold / tree.
- Lines 239-251: As someone not as familiar with how pentesting works, I think a bit more clarification of how this works would’ve helped. I ended up ChatGPT’ing several of the phrases (e.g. what does a “machine” refer to in this setting). Concretely, is a machine a Docker image / VM / some other kind of infra?
- Table 1: Minor thought, I think it might be helpful instead to highlight, in green, the top scores + fewest number of queries for each row. The highlighting of multiple “No. Queries” as green was a bit confusing initially. (so for example row 1 would highlight all the 6/6, and just 31. Row 2 would highlight 12/13, and just 49.
- Line 287: Nit, “In sum” => “In summary”

Suggested citations:
- Cybench: https://arxiv.org/abs/2408.08926
- InterCode-CTF: https://openreview.net/pdf?id=KOZwk7BFc3

**Reasons To Accept:**

- The problem is well motivated and well scoped, as the authors begin the paper by justifying the general need for good pen. testing, then focus on the limitations in using LMs and LM agents for cybersecurity, mainly due to struggles in planning.
- The method is well explained. I like Figure 1, and I appreciate how the discussion in Section 2 also grounds their improvements in comparisons with the main, prior baseline.
- Demonstrates strong improvements over the prior baseline, as shown in Table 1. What’s also great is that the approach put forth by the authors is much more efficient in terms of the number of steps taken. In addition to providing empirical evidence, the authors also include explanations for why these gaps occur (e.g. having a structured workflow enables a better representation of the Pen. Testing Tree during the problem solving process).
- I liked how the authors highlighted unproductive phenomena in the baseline method that were then mitigated or resolved by their approach (e.g. Lines 327-332).
- The authors discuss concrete limitations that are well grounded in their work, then include clear calls to action, making it easy for interested practitioners to follow up on their work.

**Reasons To Reject:**

- I am curious if there are more baselines worth comparing to. While the head to head contest with Deng et. al 2024 was carried out well, are there other baselines that should be evaluated as well? The related works seem to suggest that there are a couple more relevant works as well, but it is possible that they aren’t relevant. Perhaps some additional clarification for why other prior works are not compared against may be helpful.

---

> ### Author Response · Authors · 2025-05-30
> **Rebuttal by Authors**
>
> ## Response to Reviewer Questions:
>
> **Q1: Figure 1: What are the dotted arrows suggesting?**
>
> The referenced dotted arrows represent an interaction between LLM and the structured task tree (STT) used to guide reasoning. This clarification will be added to the caption of Figure 1.
>
> **Q2: Figure 1: Curious, in the structure you’ve provided, is it possible to “rollback” a task? My interpretation is “no” - if that’s correct, would there be a good reason to incorporate “undoing”?**
>
> No, currently we do not consider the possibility of a task “rollback”. However, this raises an interesting point. If a scenario arises where additional information is discovered about the system (e.g., a new port is identified), it is possible that a previously failing task could now succeed with this new information. While the current version of the agent does not handle this case and it never arose in our benchmark HTB challenges, this is something that we will explore further in the future. Thank you for pointing this out to us.
>
> **Q3: How comprehensive are these tasks in terms of its coverage of penetration testing settings in general? Is there potentially a pentest scenario where new tasks must be added for successful completion?**
>
> MITRE ATT&CK is a widely adopted taxonomy for adversary behavior that is updated several times per year. The maintainers explicitly note that the list is descriptive, not exhaustive, and novel techniques (e.g., previously unseen hardware faults) can be discovered between releases. In this case, a new task may need to be added for successful completion. However, most ethical penetration testing is scoped to a documented threat model where the attacker can use only certain tactics/techniques. For this form of bounded auditing, the latest form of MITRE ATT&CK would be functionally comprehensive, providing enough coverage to capture known and realistic attacker actions.
>
> **Q4: Is there a tradeoff between the structure provided by your approach vs. the freedom of Deng et al. (2024)? In the long run, do you think an LM’s planning capabilities will be good enough that it is preferable to give the LM more freedom, rather than having it subscribe to a particular task scaffold / tree.**
>
> We believe there is an inherent trade-off here. Our structured reasoning approach currently produces the highest success rates because it constrains the search space and grounds each step in MITRE ATT&CK. This not only guides the agent, but also provides a shared vocabulary that allows LLMs to better associate their internal knowledge with concrete penetration testing tasks and actions. However, as noted in the previous question, this constrained search space is not comprehensive. Techniques outside of MITRE ATT&CK cannot be used by our structured approach. As LM planning capabilities improve, agents with more freedom can exploit novel techniques (e.g., zero-days), allowing them to achieve a higher success rate in the long term. However, we foresee two scenarios where structured reasoning remains advantageous: 1) when using smaller LMs, to avoid the circular reasoning we observed; and 2) in an audit of a specific threat model or for safety-critical testing, to bound attacker actions.
>
> **Q5: Lines 239-251: As someone not as familiar with how pentesting works, I think a bit more clarification of how this works would’ve helped. I ended up ChatGPT’ing several of the phrases (e.g. what does a “machine” refer to in this setting). Concretely, is a machine a Docker image / VM / some other kind of infra?**
>
> Each machine in our evaluation is a virtualized environment hosted by HackTheBox (HTB), typically implemented as a standalone Linux/Windows VM with common services exposed. Based on the reviewer feedback, we agree that we should provide more background on penetration testing and the HTB benchmarks. We plan to incorporate a subsection, after Section 4.1, to provide more details on what exactly the benchmark machines are, their format, and what a successful penetration test would mean in a HTB machine.
>
> **Q6: Suggested citations (Cybench and InterCode-CTF)**
>
> Thank you for pointing out these relevant works. We will include these additional citations with discussion.
>
> **Minor Comments:**
>
> * **Minor nit: Perhaps you want to be using the “citep” instead of “cite” command for your citations, as that will add parentheses around the citations.**
>
> * **Table 1: Minor thought, I think it might be helpful instead to highlight, in green, the top scores + fewest number of queries for each row. The highlighting of multiple “No. Queries” as green was a bit confusing initially. (so for example row 1 would highlight all the 6/6, and just 31. Row 2 would highlight 12/13, and just 49.**
>
> * **Line 287: Nit, “In sum” => “In summary”**
>
> We agree with each of these comments and will make these changes in the camera-ready, if accepted. Thank you for taking the time to identify and write these out for us.

---

> > ### Author Response · Authors · 2025-05-30
> > **Rebuttal by Authors**
> >
> > ## Response to Review:
> >
> > **C1: Are there other baselines that should be evaluated as well?**
> >
> > To serve as a baseline for comparative evaluation, we relied on two selection criteria. Namely, the agent must be 1) model agnostic and 2) open-source. The other considered related works, namely [1, 2], are not open-source. Hence, we could not provide a scientifically fair/direct comparison for these other related works. Additionally, PentestGPT is recent and has received a large amount of attention from the research community. We believe it constitutes the current state-of-the-art penetration testing LLM agent. Prompted by reviewer comments, we performed additional literature surveys and identified another related work [3]. Similarly, this work cannot be used to provide a fair/direct comparison as it proposes a new fine-tuned model for penetration testing tasks, which would prohibit any direct comparison. We will add this additional related work with corresponding discussion to the manuscript. We appreciate the reviewer’s suggestion, which motivated us to reflect on such related directions.
> >
> > ## References:
> >
> > [1] Huang, J., & Zhu, Q. (2023, November). Penheal: A two-stage llm framework for automated pentesting and optimal remediation. In Proceedings of the Workshop on Autonomous Cybersecurity (pp. 11-22).
> >
> > [2] Happe, A., & Cito, J. (2023, November). Getting pwn’d by ai: Penetration testing with large language models. In Proceedings of the 31st ACM Joint European Software Engineering Conference and Symposium on the Foundations of Software Engineering (pp. 2082-2086).
> >
> > [3] Pratama, D., Suryanto, N., Adiputra, A. A., Le, T. T. H., Kadiptya, A. Y., Iqbal, M., & Kim, H. (2024). Cipher: Cybersecurity intelligent penetration-testing helper for ethical researcher. Sensors, 24(21), 6878.

---

> > > ### Comment · Reviewer_j49b · 2025-05-31
> > > **Response to Authors**
> > >
> > > Thanks to the authors for the comprehensive responses to my questions. I do not have any more follow ups, and will maintain my rating of 7.

---

### Official Review · Reviewer_dWYt · 2025-05-13

**Rating:** 7
**Confidence:** 4
**Ethics Flag:** 1

**Summary:**

Large Language Models have demonstrated potential in automating penetration testing. However, existing approaches often rely on the LLM's self-guided reasoning to determine next steps and specific actions. To address this limitation, this paper proposes a novel approach that leverages the Structured Task Tree, built from the MITRE ATT&CK Matrix, to guide the LLM's reasoning process. In this methodology, the LLM is provided with the STT where each node represents a tactic or technique, containing its name, a detailed description with examples, and predefined next possible tasks part which represent connections to subsequent actions. This structure enables the LLM to perform attack procedures associated with the current node, record its status and findings, and then select the subsequent node from the provided list of next possible tasks. The efficacy of this approach was evaluated on 103 discrete subtasks across 10 HackTheBox environments utilizing three LLMs. The results indicate that the proposed method significantly improves subtask completion rates for smaller models in penetration testing and also offers efficiency benefits for larger models, specifically through a reduction in the number of LLM queries.
Evaluation of Quality, Clarity, Originality, and Significance: The proposed method exhibits some originality in its application to LLM-driven penetration testing. However, there are some concerns regarding the experimental design and comparative analysis. For instance, other contemporary methods such as AutoAttacker and PentestAgent also employ LLMs for penetration testing. It is noted that AutoAttacker also reportedly utilizes knowledge from the MITRE ATT&CK Matrix, potentially augmented with additional Experience knowledge. A comparative analysis against these recent approaches would significantly strengthen the paper's contributions and contextualize its significance.

**Questions To Authors:**

On average, how many tokens does the proposed STT-based method consume per task during your evaluation? Also given that input token length can affect LLM inference speed, can you provide data on the average time taken for each task in your evaluation?
How is the STT generated? Can its structure be further modified to improve performance? I noticed that each node tends to have a similar number of possible next tasks, along with some loops. Would it be possible to design the nodes with more flexible numbers of next steps?

**Reasons To Accept:**

The evaluation effectively demonstrates the validity of guiding LLMs in penetration testing.

**Reasons To Reject:**

The experimental comparison lacks a broader comparative analysis against other recently proposed methods.
While LLM efficiency is evaluated based on the number of model queries, the analysis does not account for total token consumption (input + output). In addition, statistics on average input token counts per task are not reported. Since the proposed system introduces extensive background knowledge in each query, it may result in higher input token usage per query compared to the baseline.

---

> ### Author Response · Authors · 2025-05-30
> **Rebuttal by Authors**
>
> ## Response to Reviewer Questions:
>
> **Q1: On average, how many tokens does the proposed STT-based method consume per task during your evaluation?**
>
> On average, our method consumes approximately 324 tokens per task, compared to 321.1 tokens for the baseline (PentestGPT). In both cases, the prompt includes a predefined system prompt and dynamic content based on the task.
>
> For our method, the dynamic portion of the prompt includes either:
> 1. the description of the current task from the STT along with information provided by the tester, or
> 2. descriptions of potential next tasks from the STT when selecting the next step.
>
> PentestGPT uses a similar structure, where the dynamic content includes either:
> 1. information provided by the tester, or
> 2. the current LLM-generated PTT state, which serves as context for downstream task selection and command generation.
>
> Both approaches provide the LLM with comparable amounts of context, either in the form of task descriptions from MITRE ATT&CK (ours), or the PTT state generated through self-reasoning (PentestGPT). Hence, the actual token count is close because similar information is provided to the LLM in either case. We agree that including this information adds context to the work and our approach. This data will be added with discussion to Table 1.
>
> **Q2: Also given that input token length can affect LLM inference speed, can you provide data on the average time taken for each task in your evaluation?**
>
> We agree that inference time provides an important metric. This could be done if models are executed locally. However, the models used in this work were executed on cloud resources that we do not directly control. As such, inference time cannot be directly measured and is prone to noise due to variance in utilization, latency, etc. of the servers. For this reason, we chose to provide the number of inferences performed as our primary efficiency metric, rather than time for inference. This comment still raises an important point regarding the tokens in input prompts relative to the baseline. The average prompt length, in tokens, for our proposed approach is 324 tokens compared to 321.1 tokens for the baseline (PentestGPT). These values are close because similar information is provided to the LLM in either case. The primary difference is whether the task information is provided by the STT (ours) or the LLM-generated PTT (PentestGPT), which is prone to circular reasoning. We plan to add this new data and corresponding discussion to our evaluation in Section 4.2.2 and Table 1. We appreciate this comment as it exposes an interesting data point for our work that we did not consider initially.
>
> **Q3: How is the STT generated?**
>
> The STT is generated using systematization efforts focused on composing MITRE ATT&CK into an ordered representation to reflect how an adversary can progress through an end-to-end cyberattack (i.e., a kill-chain). These efforts use the requirements and information generated by each MITRE ATT&CK technique to represent which techniques can be completed at a given time. The allowable next steps in each case are determined by the information/context generated by the current step in the matrix. For example, in order to do “Search Victim-Owned Websites”, you must first identify an open HTTP port, which requires a prior technique, such as “Active Scanning”. Hence, dependencies exist between different techniques based on the context available. We used [1] to create the ordering of different TTPs in our STT, however, there are a number of other examples that are functionally equivalent (e.g., [2]). We will update Section 3.2 of the manuscript to discuss more details regarding how the STT is constructed/ordered.
>
> **Q4: Can the structure of the STT be modified to improve performance? Would it be possible to design the nodes with more flexible numbers of next steps?**
>
> Existing penetration testing agents using LLMs rely on self-guided reasoning, which has been shown to be prone to hallucinations and circular reasoning. We address this specific research gap by leveraging the extensive systematization efforts focused on attacker reasoning during penetration testing to construct an STT-based reasoning pipeline, which reduces the hallucination and circular reasoning present in prior work. We do believe the structure of the STT can be modified to improve performance (e.g., by adding flexibility). In fact, we are considering adding probability based node selection as an extension of our work. We appreciate and resonate with the reviewer on the possibility to improve the STT.

---

> > ### Author Response · Authors · 2025-05-30
> > **Rebuttal by Authors**
> >
> > ## Response to Review:
> >
> > **C1: The experimental comparison lacks a broader comparative analysis against other recently proposed methods.**
> >
> > PentestGPT was selected for comparative analysis because the implementation provided by the authors met two criteria: 1) model agnostic; 2) open-source. The other considered related works, namely [3, 4], are not open-source. Hence, we could not provide a scientifically fair/direct comparison for these other related works. Additionally, PentestGPT is recent and has received a large amount of attention from the research community. We believe it constitutes the current state-of-the-art penetration testing LLM agent. Prompted by reviewer comments, we performed additional literature surveys and identified another related work [5]. Similarly, this work cannot be used to provide a fair/direct comparison as it proposes a new fine-tuned model for penetration testing tasks, which would prohibit any direct comparison. We will add this additional related work with corresponding discussion to the manuscript. We appreciate the reviewer’s suggestion, which motivated us to reflect on such related directions.
> >
> > **C2: While LLM efficiency is evaluated based on the number of model queries, the analysis does not account for total token consumption (input + output). In addition, statistics on average input token counts per task are not reported. Since the proposed system introduces extensive background knowledge in each query, it may result in higher input token usage per query compared to the baseline.**
> >
> > As noted in Q1, the relative input token consumption is quite close to the baseline (324 vs  321.1  tokens per task). This is because the baseline model provides similar context in both cases. The primary difference is that this context is drawn from the PTT, generated through self-reasoning of the LLM, or the deterministic STT we have developed. We appreciate this comment as it identifies an area to improve the clarity of the manuscript. We will report this data and add additional discussion in Section 4.2.2 making it clear that our STT-based reasoning and the LLM-based baseline provide a similar number of tokens as input and why this is the case.
> >
> > ## References:
> >
> > [1] Goohs, J., Dykstra, J., Melaragno, A., & Casey, W. (2024, October). A Game Theory for Resource-Constrained Tactical Cyber Operations. In MILCOM 2024-2024 IEEE Military Communications Conference (MILCOM) (pp. 1052-1057). IEEE.
> >
> > [2] Sadlek, L., Čeleda, P., & Tovarňák, D. (2022, April). Identification of attack paths using kill chain and attack graphs. In NOMS 2022-2022 IEEE/IFIP Network Operations and Management Symposium (pp. 1-6). IEEE.
> >
> > [3] Huang, J., & Zhu, Q. (2023, November). Penheal: A two-stage llm framework for automated pentesting and optimal remediation. In Proceedings of the Workshop on Autonomous Cybersecurity (pp. 11-22).
> >
> > [4] Happe, A., & Cito, J. (2023, November). Getting pwn’d by ai: Penetration testing with large language models. In Proceedings of the 31st ACM Joint European Software Engineering Conference and Symposium on the Foundations of Software Engineering (pp. 2082-2086).
> >
> > [5] Pratama, D., Suryanto, N., Adiputra, A. A., Le, T. T. H., Kadiptya, A. Y., Iqbal, M., & Kim, H. (2024). Cipher: Cybersecurity intelligent penetration-testing helper for ethical researcher. Sensors, 24(21), 6878.

---

> > > ### Comment · Reviewer_dWYt · 2025-06-09
> > >
> > > Thank you for the detailed response and for addressing my questions. I will update my rating to 7.
> > > A discussion of more models, such as reasoning models like o3 or r1, would make it even better.

---

> > > > ### Author Response · Authors · 2025-06-10
> > > > **Rebuttal by Authors**
> > > >
> > > > We will incorporate a discussion of additional models and expected performance, particularly as model reasoning capabilities improve, in Section 4.2.3. The reviews provided a lot of good feedback that we feel will improve the work. Thank you for your time and effort in providing us with this feedback.

---

### Official Review · Reviewer_kkHX · 2025-05-23

**Rating:** 7
**Confidence:** 5
**Ethics Flag:** 1

**Summary:**

This work presents a novel guided reasoning pipeline for an LLM-based automated penetration testing agent. In contrast to previous approaches, such as PentestGPT—which relies on self-guided reasoning (Pentesting Task Tree) to create a high-level penetration testing plan—the proposed method introduces a Structured Task Tree (STT) based on the MITRE ATT&CK framework. This STT constrains the LLM’s reasoning process through explicitly defined tactics, techniques, and procedures.
Experiments conducted on 10 HackTheBox machines using three different LLMs (Llama-3-8B, Gemini-1.5, and GPT-4) demonstrate significant improvements in both task completion rates and query efficiency, particularly for smaller models like Llama-3-8B and Gemini-1.5.
Overall, the paper addresses a well-motivated and relevant challenge in the domain of LLM-based AutoPT and presents a compelling, reproducible, and practical solution. The contributions are clearly stated, methodologically sound, and empirically validated. However, the writing is confusing, and the methodology section lacks sufficient detail. Many parts are vague and difficult to understand. Furthermore, the experimental benchmarks are not clearly defined.

**Questions To Authors:**

How does it guarantee that the LLM agent will generate “executable commands (pp. 5,, line 223) in all tasks/subtasks?

Is the “in-progress” status necessary? If a task is not completed in one iteration, it is marked as failed. The paper does not mention whether multiple attempts are allowed for a single task in Section 3.

**Reasons To Accept:**

Well-motivated research with clear problem identification.
Use of real-world benchmarks and reproducible evaluation.
Strong performance improvements, especially for smaller LLMs.

**Reasons To Reject:**

Some of the key details are not completely clear. For example, it was unclear how TTPs, which are non-ordered in the MITRE ATT&CK, were processed into the ordered tasks in the proposed STT. This seems to be a non-trivial manual process that determines how effective the work can generalize to daily pentest tasks.
The definition of subtask was not clearly given.
In Section 3, the structure of the STT is not clearly explained, and how the relationship between tactics and procedures is mapped to the STT was also unclear.

---

> ### Author Response · Authors · 2025-05-30
> **Rebuttal by Authors**
>
> ## Response to Reviewer Questions:
>
> **Q1: How does one guarantee that the LLM agent will generate “executable commands”?**
>
> This and the following question appear to have identified an area where we did not provide sufficient explanation. We cannot guarantee that the LLM agent generates “executable commands”. When invalid commands were generated by the agent and applied to the HTB machine, an error is usually generated. When this error is given to the agent, the agent recognizes this and instead of marking success/failure, marks the task as “in-progress” and generates another command to run. Upon five invalid commands, the STT would mark the task as failed. We note that this is similar to the approach used by prior work, such as [1]. Section 3.1 will be updated to provide more clarification of this behavior.
>
> **Q2: Is the “in-progress” status necessary? If a task is not completed in one iteration, it is marked as failed. The paper does not mention whether multiple attempts are allowed for a single task in Section 3.**
>
> Yes, the “in-progress” status captured when the agent generated invalid commands, prompting multiple attempts. We agree that this needs additional clarification. We will update Section 3.1 to make the need and behavior of the “in-progress” status more clear.
>
> ## Response to Review:
>
> **C1: It was unclear how TTPs, which are non-ordered in the MITRE ATT&CK, were processed into the ordered tasks in the proposed STT.**
>
> Thank you for pointing this out. While ordering is not explicitly contained within MITRE ATT&CK, there exist systematization efforts focused on composing MITRE ATT&CK into an ordered representation to reflect how an adversary can progress through an end-to-end cyberattack (i.e., a kill-chain). These efforts use the requirements and information generated by each MITRE ATT&CK technique to represent which techniques are able to be completed with the goal of modeling end-to-end cyberattacks. We used [2] to create the ordering of different TTPs in our STT, however, there are a number of other examples that are functionally equivalent (e.g., [3]). We will update Section 3.2 of the manuscript to discuss how the TTPs are ordered in the STT.
>
> ## References:
>
> [1] Deng, G., Liu, Y., Mayoral-Vilches, V., Liu, P., Li, Y., Xu, Y., ... & Rass, S. (2023). Pentestgpt: An llm-empowered automatic penetration testing tool. arXiv preprint arXiv:2308.06782.
>
> [2] Goohs, J., Dykstra, J., Melaragno, A., & Casey, W. (2024, October). A Game Theory for Resource-Constrained Tactical Cyber Operations. In MILCOM 2024-2024 IEEE Military Communications Conference (MILCOM) (pp. 1052-1057). IEEE.
>
> [3] Sadlek, L., Čeleda, P., & Tovarňák, D. (2022, April). Identification of attack paths using kill chain and attack graphs. In NOMS 2022-2022 IEEE/IFIP Network Operations and Management Symposium (pp. 1-6). IEEE.

---

> > ### Comment · Reviewer_kkHX · 2025-06-09
> > **Response to the Authors: Thanks**
> >
> > Thanks for the response and the revision. The reviewer is happy with the revision, though it would be helpful if the revision could further address the ordering of TTPs as an open question.
> >
> > While the additional reference [2] can be a good source, it shall be noted that APTs are dynamic and hardly stick to one order; also, executions of TTPs are not linear: there can be branches where the executions of TTPs may appear one after another, but are in fact under different parallel sequences.
> >
> > While the paper does not address this specifically, it should be noted in open discussions or future works so that other researchers can build upon this paper more effectively.
> >
> > Nonetheless, the reviewer will maintain the same rating of 7: Good paper, accept. Congratulations and look forward to the presentation.

---

> > > ### Author Response · Authors · 2025-06-09
> > > **Rebuttal by Authors**
> > >
> > > Thank you for the feedback. We will plan to add a short discussion to Section 4.2.4 noting that executions of TTPs are not linear and under different parallel sequences. The reviews provided a lot of good feedback that we feel will strengthen the work. Thank you for your time and effort in providing us with this feedback.

---

### Decision · Program_Chairs · 2025-07-08

**Decision:**

Accept

**Comment:**

The paper proposes a reasoning pipeline for LLM-based penetration testing agents using a structured task tree derived from cybersecurity kill chains. The experiments show that the proposed approach improves subtask completion and reduces query counts.

The reviewers agree on the following points:
- The paper addresses a well-motivated and relevant challenge in automating penetration testing with LLMs.
- The proposed method demonstrates significant improvements in subtask completion rates and query efficiency, especially for smaller LLMs, and provides thorough empirical validation over strong baselines.

However, the following concerns were advised to address in the revision:
- Key aspects of the methodology, such as how unordered TTPs from MITRE ATT&CK are converted to ordered tasks in the structured task tree, and the mapping of tactics and procedures, are not clearly explained.
- The analysis does not fully account for total token consumption or inference speed, and statistics on average input token counts per task are not reported, raising questions about practical efficiency.